# False Contraindications for Vaccinations Result in Sub-Optimal Vaccination Coverage in Quito, Ecuador: A Cross-Sectional Study

**DOI:** 10.3390/vaccines11010060

**Published:** 2022-12-27

**Authors:** Felipe Andrade-Guerrero, Adriana Tapia, Vinicio Andrade, Jorge Vásconez-González, José Andrade-Guerrero, Carlos Noroña-Calvachi, Juan S. Izquierdo-Condoy, Justin Yeager, Esteban Ortiz-Prado

**Affiliations:** 1One Health Research Group, Faculty of Medicine, Universidad de las Américas, Quito 170137, Ecuador; 2Department of Pediatrics, Metropolitano Hospital, Quito 170137, Ecuador; 3Department of Pediatrics, La Paz University Hospital, 28046 Madrid, Spain; 4CDNC Consultancy Services, Quito 170137, Ecuador; 5Health Management and Research Area, Universidad Internacional Iberoamericana, Arecibo 00613, Puerto Rico; 6Grupo de Investigación en Medio Ambiente y Salud BIOMAS, Quito 170137, Ecuador

**Keywords:** health personnel, vaccines, contraindications, knowledge

## Abstract

Vaccination coverage in Ecuador has decreased since 2013, falling short of the World Health Organization’s vaccination goal. There are several causes for this deficiency in coverage, one of these are lost vaccination opportunities, which are caused when a patient without contraindications postpones, or for other reasons fails to receive a recommended immunization. The objective of this study was to determine the state of knowledge regarding vaccination contraindications among the Metropolitan District of Quito health personnel to assess missed vaccination opportunities. Through this cross-sectional descriptive study, health personnel were surveyed online and asked 18 clinical scenarios which were created to evaluate their knowledge of the true contraindications of vaccination, and measure missed opportunities. A total of 273 surveys were collected; 74% belonged to the public health system, and the rest represented by private practitioners. Of those surveyed, 98.2% of health personnel had improperly denied vaccination at least once. We specifically found vaccinations were incorrectly denied more frequently in cases where the hypothetical patient presented mild or moderate fever cases. The use of corticosteroids, autoimmune diseases, and egg allergy were also incorrectly denied (89%, 71.4%, 72.9%, and 58.6%, respectively). Among the health personnel surveyed, there is an apparent lack of knowledge of the true contraindications of vaccination and differences in knowledge about contraindications according to personnel in charge of administering immunization to children. Our preliminary results suggest that lack of education related to side effects could be biasing medical professionals’ decisions, causing them to unnecessarily delay or deny vaccinations, which likely contributes to explaining low overall vaccination coverage in Quito, the capital city of Ecuador.

## 1. Introduction

The main objective of vaccination is to promote an efficient protective immune response against a targeted pathogen to reduce the risk of developing the disease or its complications [1,2]. Various vaccines can prevent over 30 infectious diseases with excellent safety profiles and producing effective and robust immunogenicity [3]. Together with water purification, vaccines have been considered one of the best strategies to reduce morbidity and mortality, and are even more effective than antibiotics [3,4] for immune-preventable diseases.

In the last decade, the World Health Organization (WHO), the Pan American Health Organization (PAHO) other entities responsible for public health have vaccinated more than 1 billion children, annually preventing 2 to 3 million deaths. However, despite efforts on national and international levels, it is estimated that 19.7 million children under one year of age have not received essential vaccines administered during this period of time [5,6], and at least 1.5 million children under five years old die annually because of immune-preventable diseases due to lack of access to fundamental childhood vaccines [7].

Ecuador has experienced difficulties in attempting to comply with the vaccine global action plan established in 2020, whose main objective was to reach national coverage of 90% [6]. Not only was this goal missed in Ecuador, but according to the evaluation of the immunization strategy of the Ministry of Public Health (MSP, 2017), the proportion of the 2016 vaccinated population is actually lower when compared to 2013 for all the vaccines presented in this Table 1. Furthermore, the data for 2019 were not encouraging, showing stagnation, and even further decreases in some specific vaccines (Table 1) [6,8,9,10]. These trends are worrying, and it remains unresolved why vaccination rate continues to drop.

Historically, in the capital city of Quito, Ecuador, childhood vaccination coverage has never surpassed 80%; the last report for 2018 describes coverage up to 79% [9,11]. Missed vaccination opportunities appear to significantly contribute to these low values, which occur in instances when a child who is eligible for vaccination, and has no contraindications, visits a healthcare service and does not receive all the recommended vaccine doses [12]. Health caregivers’ refusal to vaccinate, or inadequate or absent advice from health personnel contribute to the problem of missed vaccinations [13,14]. Even rescheduling or postponing vaccinations can result in a lost opportunity to immunize, due to geographical distances between patients and healthcare providers, exasperated by insufficient transport availability, resulting in incomplete immunization schedules [15].

Health personnel oversee administering vaccines and resolving questions or uncertainties related to immunizations within the community. A misinterpretation of vaccines’ indications and contraindications can lead to missed immunization opportunities or unnecessary delays [16,17]. Our objective was therefore to understand the current state of physicians’ understanding related to vaccines, in order to determine if the low vaccination rates in the Metropolitan District of Quito could be attributable to unnecessarily missed opportunities, rather than other impediments, such as the inability of patients to reach vaccination points. A more thorough assessment of physicians’ knowledge related to the true contraindications of vaccinations in the capital city is an essential first step if there is to be hope of increasing vaccination coverage of the Ecuadorian population.

## 2. Materials and Methods

### 2.1. Study Design

We present a preliminary descriptive, observational, cross-sectional study to meet the proposed objectives. The study was conducted from September to December 2020 in the Metropolitan District of Quito (Quito canton). Participants responded to an online questionnaire sent through all available institutional mails for health personnel from public and private entities. 

This questionnaire was sent to the health personnel (specialist doctors, general practitioners, and nursing staff) of primary public health care and the pediatric doctors affiliated with the Ecuadorian society of pediatrics affiliate Pichincha. In total, 626 surveys were sent throw institutional mail, of which 275 replies were obtained, with 273 voluntarily agreed to conduct this questionnaire, and two replies declining participation, leading to a response rate of 44%.

### 2.2. Data/Measurement Sources

A questionnaire was developed based on the one used by Rivero et al., which tested different scenarios of actual contraindication vaccination knowledge. We created 18 scenarios adapted to Ecuador’s vaccination scheme [18], which was prepared electronically in the Office 365 program Microsoft Forms©. The survey consisted of scenarios that tested the knowledge of health personnel about vaccination contraindications where each scenario had three response options: vaccinate, do not, or postpone vaccination. All the items in the survey did not constitute a contraindication to vaccination.

### 2.3. Control of Sources of Bias

Due to the pandemic, primary care personnel shifted employment and subsequently and institutional address, hindering our ability to personally distribute the survey, which could have resulted in a lower response rate than expected.

### 2.4. Study Size

The sample size was not calculated since it was a non-probabilistic sample, which included all the participants who voluntarily agreed to take the questionnaire.

### 2.5. Statistical Methods

The data were managed through the statistical program R studio version 1.4.1106, through which it was tabulated, and descriptive statistical analyzes of central tendency and frequency analysis were performed. The ggplot2 package of the R studio program was used for the exploratory study.

## 3. Results

Replies were obtained from 275 participants, of which 273 (99.3%) voluntarily agreed to carry out this questionnaire, and 2 (0.7%) replied that they declined to participate. Of the participants who answered the survey, 202 (74%) belonged to the public health network and 71 (26%) to the private service.

Specialist doctors predominated the responses of those who agreed to participate in the survey (34.4%, *n* = 94). Among those, 62 belonged to private healthcare facilities and 32 to the public healthcare system. Nursing personnel constituted 31.2% (*n* = 85) of replies and were limited to members of the public healthcare system. General practitioners 24.9%, (*n* = 68) were distributed in the public system (*n* = 60) and in the private system (*n* = 8). Finally, primary health care technicians 9.5% (*n* = 26).

Nearly all medical professionals surveyed (98.2%, *n* = 268) had denied vaccination at least once in the questionnaire. Surprisingly, among all participants only five respondents (1.8%) answered the entire questionnaire correctly, vaccinating in all presented scenarios, indicating high rates of incorrect rejections for possible vaccinations. 

Of the scenarios we proposed in the survey, many respondents responded that they would delay vaccination in children with fever (89%, *n* = 243) (Table 2). Regarding the use of drugs, over half of the respondents decided to vaccinate children receiving antibiotic treatment, 52.4% (*n* = 143). Conversely, 71.4% (*n* = 195) indicated they would not recommend immunizing a child who was on steroid therapy.

Health personnel decided to deny vaccination (72.9% *n* = 199) in children with autoimmune diseases such as agammaglobulinemia, for vaccinations against chickenpox. Similarly, a slight majority of the respondents (58.6%, *n* = 160) declined vaccination of the triple viral (measles, mumps, and rubella) for patients with egg allergies. Further specific instances are illustrated in Table 2 and Figure 1.

## 4. Discussion

Childhood vaccination coverage in Ecuador has lagged both national and international goals for 2020. Rectifying any instances where vaccinations are improperly rejected could assist in reducing children’s morbidity and mortality. Vaccines are estimated to prevent almost six million deaths/per year and to save 386 million life years and 96 million disability-adjusted life years (DALYs) globally [19].

One of the leading causes for Ecuador’s failure to achieve the desired vaccination coverage could be due to missed vaccination opportunities. These missed opportunities often occur when a patient without contraindications postpones, or never receives the corresponding immunization. This problem is not novel, in fact it has been described frequently in low- and middle-income countries [15]. Several reviews suggest that when health personnel misinterpret a vaccines contraindication, these errors could lead to improper postponement or vaccination denial [20,21]. According to Tampi et al., missed vaccination opportunities can vary greatly, but total from 5% to 37% of vaccination opportunities in Latin America [22]. Gaps in the healthcare provider’s knowledge, as well as their attitudes and behaviors related to vaccine contraindications, and patient communication skills with diverse healthcare teams are thought to be responsible for these reductions in vaccination opportunities [20,22]. 

Jimbo-Sotomayor et al. reported a study on 368 children from Quito, illustrating that 33.4% had an incomplete vaccination scheme and only 44.7% of those able of being vaccinated were indeed immunized [23]. The actors responsible for not vaccinating these youth were attributed to the parent or caregiver (76.2% of cases) as well as health personnel (19%). Although parents/caregivers in this study represent a larger proportion of decision-makers. Yet these missed opportunities for vaccinations equally can occur when healthcare professionals provide misinformation to caregivers or parents, as well as when physicians or medical professionals erroneously deny or postpone vaccinations [13,15,23]. Therefore, understanding under which circumstances healthcare professionals inaccurately reject or delay is vital to increasing vaccination rates. 

Mild fevers were one of the main scenarios which were regularly misinterpreted by health personnel as a contraindication for immunization. It is common for health personnel to postpone vaccination when a patient presents with a mild fever (≦38 °C) or in case of a mild infection that present a mild fever, despite the fact both scenarios there is no evidence to consider it as a contraindication for vaccination [21,24,25]. This misunderstanding is not unique to Latin America, it has been also reported by Rivero et al. where 77% of European healthcare providers delayed or refused vaccination due to mild fever. In our study, a similar percentage (69.6% of health workers) recommended postponing vaccination, and 19.4% considered not vaccinating a patient with a mild fever. Postponing vaccination may be justified if the patient has a serious disease (e.g., severe infection), if the objective is to avoid incorrectly attributing fever symptoms to the vaccine; however, it does not justify denying vaccination in all cases of mild fever where a physical examination shows no indication of pathology [18,24,25].

The use of medications as a justification for not vaccinating Is a frequent stigma that health personnel must resolve; among the most common are when patients are taking antibiotics and/or steroids. It is generally accepted that the use of antibiotics has no effect on the immunogenicity of the vaccine, nor does it cause any adverse effects [24,25]. While most health workers (52.4%) in our study decided to vaccinate children taking antibiotics, 47.6% unnecessarily delayed or denied immunization, suggesting misinformation or ignorance of established guidelines, likely resulting in potentially unvaccinated patients. Like antibiotic treatments, the use of low-dose steroids is not a reason to postpone immunization [25]. Our result suggests that most health personnel similarly delay vaccination when patients are on steroid medications. 

The history of preterm labor can ”e pe’ceived as a precaution but not a contraindication when it presents associated disorders such as heart disease, bronchopulmonary dysplasia, neurological disorders, recurrent infections, apnea, and chronic pharmacological therapies [24,25]. Therefore, the scenario we presented to respondents did not contraindicate immunization. Yet 12.5% of medical professionals opted to postponed, and 60.4% of the participants contraindicated vaccination, suggesting significant ignorance of vaccine contraindications in a patient with immune alterations.

Most of the severe reactions after administration of the MMR vaccine (measles, mumps, and rubella) occur in children who are not allergic to eggs. These vaccines are produced in chicken embryo fibroblasts, not eggs [25]. Therefore, the MMR vaccine is just as safe as any other vaccine, is not contraindicated by egg allergy, and can be administered routinely without preliminary testing, like for the influenza vaccine. However, in our sample, 51.3% of the health personnel contraindicated the MMR vaccine administration [24,25,26]. Although nearly 60% of the respondents’ state that they know that immunization can be administered, a significant percentage of respondents still decided to postpone or deny vaccination, which could result in vaccination opportunity loss as Ecuador has rural and urban regions, many of them with geographical barriers that make it challenging to comply with the vaccination schedules.

In this study, most participants were health workers from primary care centers in the public system. Being a preliminary study and with a small sample of participants and from a single city in Ecuador, we acknowledge albeit likely, we cannot generalize the results at the national level. We highlight that all health personnel, especially those on the first line of care, should possess up-to-date knowledge of vaccines, regardless of profession or specialization. The specialist doctors and/or nurses applying vaccines must specifically have standardized knowledge related to specific vaccinations and their respective contraindications. It is possible that many healthcare workers may not be directly in charge of the child’s immunization, still, they are important sources of information to parents and caregivers, and can spread the disinformation related to contraindications. Together this misinformation can result in lost vaccination opportunities, resulting in suboptimal vaccination status in Quito, and perhaps nationally. On the other hand, the lack of clear practice guidelines on vaccines could cause health personnel to make individualized decisions, giving rise to significant discrepancies in the application of immunizations among medical personnel. Institutional regulations and adequate and up-to-date information on the indications, precautions, and contraindications of vaccines could help educate health personnel, which will undoubtedly contribute to increasing vaccination coverage in the population.

## 5. Limitations

We present a non-probabilistic opportunistic sample, so we cannot extrapolate to the Ecuadorian general population. As it is an online questionnaire, we cannot control the honesty at the time in which the different participants answered the questions, therefore participants would have had the opportunity to consult references before replying to scenarios.

## 6. Conclusions 

We suggest that the incomplete, or erroneous perceptions or false information given by medical staff (specialist doctors, general practitioners, and nursing staff) could be one of the main reasons why the vaccination goals have been missed in Quito. Although our study was, regional and limited in a single city, we cannot ensure results would be generalized to national level, though as many doctors are educated in universities within Quito, our results could potentially be also found more broadly in future studies. These trends of postponed or rejected vaccinations could be easily, and rapidly, rectified by properly educating medical practitioners and assuaging concerns with evidence-based medicine so that vaccines can serve their essential role as a control measure for the spread of infectious diseases.

## Figures and Tables

**Figure 1 vaccines-11-00060-f001:**
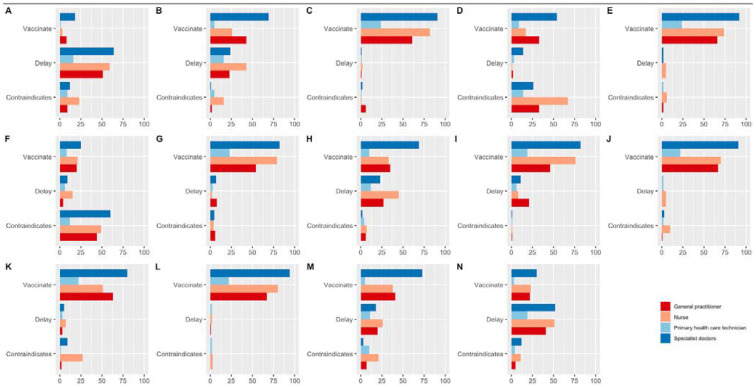
Bar graph of the knowledge of the personnel surveyed about false contraindications for vaccines, according to the level of health care to which they belong: (**A**). moderate fever, (**B**). use of antibiotic therapy, (**C**). children who are breastfeeding, (**D**). History of allergy to eggs, (**E**). History of mild allergic reaction to another vaccine, (**F**). History of autoimmune diseases such as agammaglobulinemia, (**G**). Children who live with pregnant women, (**H**). Children who are recovering from influenza, (**I**). History of prematurity, (**J**). DTP vaccine immunization in children with a family history of epilepsy, (**K**). Immunization of the DTP vaccine in children with a history of controlled epilepsy, (**L**). History of allergy to penicillin, (**M**). Vaccination in advance of the card, (**N**). Corticosteroid therapy.

**Table 1 vaccines-11-00060-t001:** Vaccination coverage (%) in Ecuador in the 2010–2019 period.

	Vaccine Period									
Vaccines	2010	2011	2012	2013	2014	2015	2016	2017	2018	2019
HepBB	5%	7%	16%	69%	79%	75%	47%	61%	70%	71%
MMR2	91%	92%	55%	83%	85%	76%	64%	73%	74%	76%
DTP3	100%	100%	100%	87%	83%	78%	83%	85%	85%	85%
PCV3	17%	71%	94%	90%	100%	81%	84%	84%	85%	83%
Polio3	100%	100%	100%	87%	84%	84%	79%	83%	85%	85%
MMR1	N/D	N/D	N/D	N/D	N/D	84%	86%	81%	83%	83%
DTP1	100%	100%	100%	87%	84%	80%	82%	84%	86%	86%
BCG	100%	100%	100%	90%	89%	88%	84%	88%	90%	86%

N/D: No data available. Adapted from World Health Organization. (2020). Vaccine coverage. HepBB: Hepatitis B Vaccine. MMR: Measles, Mumps, and Rubella. DTP: Diphtheria, tetanus, and pertussis. PCV: Pneumococcal conjugate vaccine. BCG: Bacille Calmette-Guerin.

**Table 2 vaccines-11-00060-t002:** Summary of responses from health personnel.

Case Scenario	Theme	Vaccinate % (n)	Delay % (n)	Contraindication % (n)
Fever (38 °C) with normal physical examination and mild flu symptoms in a 5-year-old child	Fever	11% (30)	69.6% (190)	19.4% (53)
Taking antibiotic therapy and going to receive immunization	Antibiotic therapy	52.4% (143)	38.8% (106)	8.8% (24)
Breastfeeding a healthy 9-month-old baby	Breastfeeding	94.5% (258)	1.8% (5)	3.7% (10)
MMR immunization with egg allergy	Egg allergy	41.4% (113)	7.3% (20)	51.3% (140)
Mild, non-anaphylactic allergic reaction prior to previous vaccination	Mild allergic reaction	93.8% (256)	2.6% (7)	3.6% (10)
History of autoimmune diseases such as agammaglobulinemia prior to varicella immunization	Autoimmune diseases	27% (74)	12.5% (34)	60.4% (165)
History of living with immunosuppressed people for rotavirus immunization	Living with an immunosuppressed person	87.2% (238)	7.3% (20)	5.5% (15)
Patient discharged for influenza, however, presents mild symptoms	Convalescence	53.8% (147)	39.2% (107)	7% (19)
2-month-old patient with a history of prematurity	Premature	81.7% (223)	16.8% (46)	1.5% (4)
Patient with a family history of epilepsy to receive DPT immunization	Family history of epilepsy	91.6% (250)	2.6% (7)	5.9% (16)
Patient with a personal history of epilepsy controlled for immunization of the DPT vaccine	Epilepsy	79.1% (216)	6.6% (18)	14.3% (39)
12-month-old patient with an allergy to penicillin	Allergy to penicillin	96.3% (263)	1.8% (5)	1.8% (5)
4-month-old patient who comes to receive the immunization days before the date according to the vaccination schedule	Advanced immunization	57.5% (157)	27.5% (75)	15% (41)
Taking corticosteroid therapy and going to receive chickenpox immunization	Corticosteroid therapy	28.6% (78)	59.7% (163)	11.7% (32)

## Data Availability

The data from which the results of this research were obtained are available for those who so wish. In case you wish to access this information, you must make a personal request to the responsible author through the following e-mail felipe5n@hotmail.com.

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
