# Peer review of "False Contraindications for Vaccinations Result in Sub-Optimal Vaccination Coverage in Quito, Ecuador: A Cross-Sectional Study"

_vaccines, 2022, doi:10.3390/vaccines11010060_

Round 1
Reviewer 1 Report
This is a preliminary study on the state of knowledge on appropriate vaccination contraindications in the Metropolitan District of Quito.
Here, as bullet points, my main concerns and comments.
1) the title should be changed because the survey was not conducted in Ecuador but in the Metropolitan District of Quito and this "regional/distric selection" might explain the relatively small numbers of participants in the study.
2) The study should be defined as a preliminary one
3) Which was the response rate to the online survey? Id est, how many people were contacted?
4) At the end of the introduction lacks a sentence that explains how the Authors decided to assess the physicians' knowledge on vaccines contraindications in the selected District. Moreover, in the text (line 116) is written that "nursing personnel..." thus stating that not only physicians were included in the survey. Responses obtained from nurses and physicians should be showed in separeted Tables because I think that the knowledge on appropriate contraindications could be different due to a different academic background.
5) I cannot find enclosed the whole questionnaire; I think it should be included as online Appendix. It should be soundly defined why these specific scenarios were selected.
6) The conclusions could be better explained.
7) The bibliography is not always sound enough
***
More in details:
lines 24-25: the sentence is not complete.
line 30: a Regional result cannot be extended to the whole Ecuador
lines 37-38: the references are inverted and not so sound
line 54: the reference to Table 1 is mixed with the citations
line 73: in my opinion the regional results cannot be generalized thus here it's not in "Ecuador" but "in The Metropolitan District of Quito"
line 81: a space lacks after "2020"
line 83: here should be described who was considered as "health personnel"
line 96: please write "did" instead of "do"
lines 98-100: the concept should be better explained; does it imply a lower response rate than expected?
lines 102-103: the sentence should be re-phrased more clearly
line 110: here lacks the total number of health personnel contacted
line 129: please insert a comma instead of the round bracket before "n=..." and add the denominator after 160
line 130: I suggest writing "are showed" instead of "can be seen"
Please revise the title of Table 2.
Figure 2 is not very clear in terms of "...according to the level of health care to which they belong.."
line 153: here should be specified that the study of Tampi et al. refers to Latin America
lines 204-209: as previously written, the results of this preliminary survey cannot be generalised and this aspect should be more stressed
lines 224-225: the limitations coud be better described
line 227: the term "medical personnel" in not precise
Author Response
Point by Point Letter To: Dr. Tamara Sipka Assistant Editor Vaccine’ Tittle: False contraindications for vaccinations result in sub-optimal vaccination coverage in Ecuador: A cross-sectional study Manuscript ID: vaccines-2023147
Dear Editor and reviewers, thank you very much for your effort, observing our manuscript, and offering us some comments intended to improve our research. We have completed the revision and we have fulfilled all your comments and suggestions. All the changes are highlighted in red within the main manuscript and highlighted in this point by point letter. We also have reviewed the entire version to fulfill with the language requirements.
Reviewer #1 Comments This is a preliminary study on the state of knowledge on appropriate vaccination contraindications in the Metropolitan District of Quito. Here, as bullet points, my main concerns and comments.
1) the title should be changed because the survey was not conducted in Ecuador but in the Metropolitan District of Quito and this "regional/distric selection" might explain the relatively small numbers of participants in the study.
We have changed the tittle accordingly
2) The study should be defined as a preliminary one
We have added a comment about this
3) Which was the response rate to the online survey? Id est, how many people were contacted?
We have added the information in the main text. We thank the reviewer of the comment.
4) At the end of the introduction lacks a sentence that explains how the Authors decided to assess the physicians' knowledge on vaccines contraindications in the selected District. Moreover, in the text (line 116) is written that "nursing personnel..." thus stating that not only physicians were included in the survey. Responses obtained from nurses and physicians should be showed in separeted Tables because I think that the knowledge on appropriate contraindications could be different due to a different academic background.
5) I cannot find enclosed the whole questionnaire; I think it should be included as online Appendix. It should be soundly defined why these specific scenarios were selected.
We thank the reviewer for his comment. We included the questionnaire in English and Spanish.
6) The conclusions could be better explained.
We have updated our conclusion section
7) The bibliography is not always sound enough
We have added new references *** More in details: lines 24-25: the sentence is not complete. We revised the main text and changed it accordingly.
line 30: a Regional result cannot be extended to the whole Ecuador
we have changed this accordingly
lines 37-38: the references are inverted and not so sound
Fixed
line 54: the reference to Table 1 is mixed with the citations
We have corrected it
line 73: in my opinion the regional results cannot be generalized thus here it's not in "Ecuador" but "in The Metropolitan District of Quito"
We added your suggestions
line 81: a space lacks after "2020"
Done
line 83: here should be described who was considered as "health personnel"
we reviewed the text and rearranged the main text, lines 89-92.
line 96: please write "did" instead of "do"
We did the correction
lines 98-100: the concept should be better explained; does it imply a lower response rate than expected?
We had explain more
lines 102-103: the sentence should be re-phrased more clearly
We did the correction
line 110: here lacks the total number of health personnel contacted
We have added the information in the main text. We thank the reviewer of the comment.
line 129: please insert a comma instead of the round bracket before "n=..." and add the denominator after 160
We did the correction
line 130: I suggest writing "are showed" instead of "can be seen"
We did the correction
Please revise the title of Table 2.
We did the correction
Figure 2 is not very clear in terms of "...according to the level of health care to which they belong.." This was amended line 153: here should be specified that the study of Tampi et al. refers to Latin America
We did the correction and specified that the study refers to Latin America
lines 204-209: as previously written, the results of this preliminary survey cannot be generalised and this aspect should be more stressed
We did the correction
lines 224-225: the limitations coud be better described
We better describe
line 227: the term "medical personnel" in not precise
We better describe
Reviewer 2 Report
Motivated by low vaccination coverage in Ecuador, a survey was conducted to get information about knowledge of healthcare personel on contraindications for vaccination. The results can be informative, but adding information of a ‘control group’ (knowledge of healthcare personel in countries with higher vaccination coverage) would increase the value of the study.
General comments:
Lack of knowledge about contraindications for vaccination could be an important reason for low vaccination coverage. However, comparing the findings of this study to comparable surveys in countries with higher coverage would be interesting. In Discussion, l. 171, such a comparison is given concerning ‘fever’. Can this be extended to other scenario’s?
Please make clear if ‘vaccination coverage’ in this paper refers to childhood vaccination, or this is broader?
Other major comments:
l. 27-28: In the Methods and Results in this paper there is nothing mentioned about investigating differences in knowledge between the different professions. If such an analysis was done, please specify method of testing and results in the main text. Otherwise remove this sentence from the abstract.
l. 73,74: ‘……. rather than other impediments, such as the inability of patients to reach vaccination points.’ This study cannot give an answer about how big is the influence of ‘lack of knowledge’ and the influence of other causes. Please be careful in suggesting ‘lack of knowledge’ is the main factor.
Methods: There should be reported to how many persons the questionnaire was sent, to get the response rate. If the exact number of persons reached by the mailing is unknown, please give a reasonable estimate.
Discussion: Please focus in the Discussion on the results of this study and their relation to other findings and literature. Text in l. 145-167 is more suitable for the Introduction.
The numbering of the references seems to be incorrect, as number 1 is missing.
Minor comments:
l. 24,25: Incomplete sentence
l. 28: not clear what is ‘combined’.
l. 44: Is the number of 19.7 million an annual number, is related to the last decade or to a recent year?
l. 50: what is meant by ‘district coverage’?
l. 52: ‘a constant decline form 2013 until 2016’ is not a correct description. Several vaccines have a higher coverage in 2014 or 2015 compared to 2013. It will be correct to say that coverage in 2016 is lower compared to 2013 for all the vaccines presented in this table.
Table 1: Please add the meaning of all abbreviations.
Maybe presenting these numbers in a plot might better show increase and decrease?
l. 54: The text ‘Table 1’ should not be between the brackets
l. 99: ‘preventing’ is probably not what is meant here, consider ‘hinder’ or ‘hamper’.
l. 114- 118: The 28 persons missing in this listing, was their profession ‘other’ or were these ‘unknown’?
l. 119: This number should be 268?
l. 128: The denominator of 275 seems to be incorrect (and can be omitted).
Table 2 and Figure 1 present exactly the same results. Please choose for one presentation.
Discussion: References Tampi et al. (2022) and Jimbo-Sotomayor et al (2019) are not in the correct style.
l. 182: For clearness, add ‘taking antibiotics’ after ‘children’.
Author Response
Reviewer #2 Comments
Motivated by low vaccination coverage in Ecuador, a survey was conducted to get information about knowledge of healthcare personel on contraindications for vaccination. The results can be informative, but adding information of a ‘control group’ (knowledge of healthcare personel in countries with higher vaccination coverage) would increase the value of the study.
General comments:
Lack of knowledge about contraindications for vaccination could be an important reason for low vaccination coverage. However, comparing the findings of this study to comparable surveys in countries with higher coverage would be interesting.
Thanks, we have expanded this within the discussion section
In Discussion, l. 171, such a comparison is given concerning ‘fever’. Can this be extended to other scenario’s?
we have expanded this within the discussion section
Please make clear if ‘vaccination coverage’ in this paper refers to childhood vaccination, or this is broader?
We had clarified and change the main text
Other major comments:
- 27-28: In the Methods and Results in this paper there is nothing mentioned about investigating differences in knowledge between the different professions. If such an analysis was done, please specify method of testing and results in the main text. Otherwise remove this sentence from the abstract.
We had erase the sentence, however we only recruit health personal in charge of administrating vaccines.
- 73,74: ‘……. rather than other impediments, such as the inability of patients to reach vaccination points.’ This study cannot give an answer about how big is the influence of ‘lack of knowledge’ and the influence of other causes. Please be careful in suggesting ‘lack of knowledge’ is the main factor.
We agree with the opinion of the reviewer. we have changed the text
Methods: There should be reported to how many persons the questionnaire was sent, to get the response rate. If the exact number of persons reached by the mailing is unknown, please give a reasonable estimate.
We did the correction and had added the response rate.
Discussion: Please focus in the Discussion on the results of this study and their relation to other findings and literature. Text in l. 145-167 is more suitable for the Introduction.
We had added new bibliography
The numbering of the references seems to be incorrect, as number 1 is missing.
We corrected it
Minor comments:
- 24,25: Incomplete sentence
- 28: not clear what is ‘combined’.
We corrected it
- 44: Is the number of 19.7 million an annual number, is related to the last decade or to a recent year?
We have clarified it
- 50: what is meant by ‘district coverage’?
We have clarified it
- 52: ‘a constant decline form 2013 until 2016’ is not a correct description. Several vaccines have a higher coverage in 2014 or 2015 compared to 2013. It will be correct to say that coverage in 2016 is lower compared to 2013 for all the vaccines presented in this table.
We agree with the reviewer and had corrected in the main text.
Table 1: Please add the meaning of all abbreviations.
Maybe presenting these numbers in a plot might better show increase and decrease?
We added the meaning
- 54: The text ‘Table 1’ should not be between the brackets
We corrected it
- 99: ‘preventing’ is probably not what is meant here, consider ‘hinder’ or ‘hamper’.
We corrected it
- 114- 118: The 28 persons missing in this listing, was their profession ‘other’ or were these ‘unknown’?
- 119: This number should be 268?
- 128: The denominator of 275 seems to be incorrect (and can be omitted).
We omitted it
Table 2 and Figure 1 present exactly the same results. Please choose for one presentation.
Discussion: References Tampi et al. (2022) and Jimbo-Sotomayor et al (2019) are not in the correct style.
We corrected it
- 182: For clearness, add ‘taking antibiotics’ after ‘children’.
We added taking antibiotics after children
Round 2
Reviewer 1 Report
There are my comments in bullet points after the revision and the changes made.
-The results and their description are not always so sound/strong.
-I think that Figure 1 is not helpful, it's difficult to interpretate at a first glance.
-I haven't found the Questionnaire and the separeted Tables suggested for evidencing/describing nurses and doctors' possibly different approach.
In conclusion, I think that the best format of this preliminary work, for this Journal, could be the Letter or the Communication one.
Author Response
Reviewer #1 Comments
There are my comments in bullet points after the revision and the changes made.
-The results and their description are not always so sound/strong.
We have improved our explanation of the results; it is highlighted in red
-I think that Figure 1 is not helpful, it's difficult to interpretate at a first glance.
We have improved our figure and now it should be more clear
-I haven't found the Questionnaire and the separated Tables suggested for evidencing/describing nurses and doctors' possibly different approach.
The questionnaire was submitted to the Vaccine MDPI submission system and should be included within this manuscript
In conclusion, I think that the best format of this preliminary work, for this Journal, could be the Letter or the Communication one.
Thanks for your comment, we respectfully disagree with you on this question. We believe the investigation deserve full consideration as original research
Reviewer 2 Report
The results in line 122 - 130 are still incomplete/unclear:
1) 94 doctors + 83 nurses + 68 GPs add up to 245. What was the profession of the other 28?
2) It seems that among the 273 responders there were 268 denying vaccination at least once and 5 denying never. Then the n = 273 following 98.2% is incorrect.
It is a missed opportunity not to report the results stratified by doctors/nurses/GPs/other.
Author Response
Reviewer #2 Comments
The results in line 122 - 130 are still incomplete/unclear:
Thanks, we have corrected all the observations
- 94 doctors + 83 nurses + 68 GPs add up to 245. What was the profession of the other 28?
We have added all the numbers correctly
- It seems that among the 273 responders there were 268 denying vaccination at least once and 5 denying never. Then the n = 273 following 98.2% is incorrect.
Thanks for pointing this out, we have corrected all the typos
It is a missed opportunity not to report the results stratified by doctors/nurses/GPs/other.
We have included the stratified results, thanks for your suggestions
Round 3
Reviewer 1 Report
Thanks for the comments.
Author Response
Dear Reviewers, please find attached the newer version of our manuscript
